# Biomarkers for Alzheimer’s Disease Early Diagnosis

**DOI:** 10.3390/jpm10030114

**Published:** 2020-09-04

**Authors:** Eva Ausó, Violeta Gómez-Vicente, Gema Esquiva

**Affiliations:** Department of Optics, Pharmacology and Anatomy, University of Alicante, 03690 Alicante, Spain; eva.auso@ua.es (E.A.); vgvicente@ua.es (V.G.-V.)

**Keywords:** Alzheimer’s disease, biomarkers, early diagnosis, biofluids

## Abstract

Alzheimer’s disease (AD) is the most common cause of dementia, affecting the central nervous system (CNS) through the accumulation of intraneuronal neurofibrillary tau tangles (NFTs) and β-amyloid plaques. By the time AD is clinically diagnosed, neuronal loss has already occurred in many brain and retinal regions. Therefore, the availability of early and reliable diagnosis markers of the disease would allow its detection and taking preventive measures to avoid neuronal loss. Current diagnostic tools in the brain, such as magnetic resonance imaging (MRI), positron emission tomography (PET) imaging, and cerebrospinal fluid (CSF) biomarkers (Aβ and tau) detection are invasive and expensive. Brain-secreted extracellular vesicles (BEVs) isolated from peripheral blood have emerged as novel strategies in the study of AD, with enormous potential as a diagnostic evaluation of therapeutics and treatment tools. In addition; similar mechanisms of neurodegeneration have been demonstrated in the brain and the eyes of AD patients. Since the eyes are more accessible than the brain, several eye tests that detect cellular and vascular changes in the retina have also been proposed as potential screening biomarkers. The aim of this study is to summarize and discuss several potential markers in the brain, eye, blood, and other accessible biofluids like saliva and urine, and correlate them with earlier diagnosis and prognosis to identify individuals with mild symptoms prior to dementia.

## 1. Introduction

### 1.1. Pathophysiology of AD and Clinical Manifestations

The lesions of Alzheimer’s disease (AD) include pathological changes in the brain such as the accumulation of proteins (amyloid-β (Aβ) peptide and Tau); the degeneration of neurons and synapses, most noticeably in the neocortex and the hippocampus, which leads to structural changes as well as to the loss of functional connectivity, and the alterations of reactive processes like neuroinflammation and plasticity, related to oxidative stress and mitochondrial dysfunction [1]. Some of these hallmarks can be detected in the prodromal stage of the disease, also referred to as mild cognitive impairment (MCI) due to AD, when the symptoms are not yet obvious.

Amyloid-β deposits are widely distributed in the brain and follow an anterograde sequence originating in five phases in which different brain regions are hierarchically involved [2,3,4]. The five phases go from phase 1, when the deposits are exclusively found in the isocortex, to phase 5, when the cerebellum and several brainstem nuclei, such as the pontine nuclei and the locus coeruleus, among others, are involved [2,4]. The progression of Tau pathology is also staggered from the transentorhinal and entorhinal cortex to the isocortex via the hippocampus, with a heterogeneous and area-specific neuronal loss [2,3,4]. It is well-established that the accumulation of Tau protein takes place specifically in neurons and occurs in their cell body as neurofibrillary tangles (NFTs), in their dendrites as neuropil threads (NT), and in their axons forming the senile plaque neuritic corona [3].

The Braak stages, based on phospho-Tau accumulation within connected brain regions, defines the progression of AD neuropathology. I–II refer to the entorhinal cortex, III–IV to the hippocampus/limbic system, and V–VI to the frontal and parietal lobes. 

### 1.2. Diagnostic Tools

The progress in the diagnosis of AD has noticeably improved with the development in the last decades of noninvasive neuroimaging techniques that allow the visualization of structures in vivo. Some examples are novel magnetic resonance imaging (MRI), metabolic changes detected by positron emission tomography (PET), and amyloid imaging. These techniques permit the detection of pre-symptomatic diagnostic biomarkers in the brains of cognitively normal elderly individuals and also serve to monitor disease progression after the onset of symptoms [1]. Due to their reliability and high discriminative capacity in the pre-dementia state, volumetric approaches of the high-resolution subfield are useful, as well as diagnostic techniques in order to study the early changes in the most affected brain structures [2,3,4]. With all these tools, the typical lesions related to protein accumulation and the structural changes in certain brain areas are easier to detect and; therefore, constitute the basis of the diagnosis.

In addition, the advancement in the past few years of omics technologies (genomics, transcriptomics, proteomics, metabolomics, secretomics, etc.) has made possible the analysis of a wide range of AD hallmarks referring to both, sporadic and familial cases. These tools facilitate the analysis of human fluid samples of diverse nature such as blood, tears, urine, or saliva, whose collection in most cases does not require trained professionals and has the advantage of being noninvasive due to easy accessibility. The importance of identifying and developing reliable and sensitive tools for the early diagnosis of AD relies on the potential benefits for the patients, including timely access to medical treatments to slow down the progression of the disease and; therefore, preservation of longer cognitive capacity, or even the possibility to plan for the future.

## 2. Invasive Biomarkers

### 2.1. Changes in Specific Brain Areas as Early Biomarkers

The locus coeruleus (LC) is a neuromelanin-rich brainstem structure thought to modulate attention and memory and is the major source of noradrenaline in the brain. In the asymptomatic stage of AD, Tau NFTs are observed in the LC [5,6] prior to their presence in other cerebral areas such as the entorhinal cortex and the neocortex [7,8,9,10]. These Tau aggregates precede typical neuronal loss in the LC during AD progression [11]. Studies using unbiased stereology have revealed an average decrease in LC volume of 8.4% for each Braak stage increment, as well as neuronal loss mainly in the rostral/middle area of the LC, progressing from 30% in the prodromal stage to 55% when dementia is diagnosed [11]. Functionally, this neuronal loss has correlated with cognitive dysfunction [12] and reduced noradrenaline levels in the hippocampus and the cortex [13]. Additionally, a two-fold increase in Tau accumulation was also observed from Braak stage 0 to I [9]. Therefore, the detection using in vivo imaging of early structural tissue modifications such as the decrease in the LC volume or metabolic changes would support the diagnosis and could potentially slow down disease progression if the patient benefits from treatments in the appropriate time [14,15].

Although there is controversy regarding the accelerated rates of brain atrophy at the preclinical stage of the disease, it seems that the medial temporal lobe and, particularly, the hippocampus are brain structures early affected by NFTs and neurite loss. Studies using voxel-based morphometry and high-resolution MRI have revealed hippocampal atrophy in AD patients’ brains at the preclinical stage, up to 10 years before the diagnosis of dementia [16], and even before MCI [17,18]. The magnitude of atrophy in the hippocampus and its subfields determines the progression to either MCI or AD [19,20,21]. Thus, studies using radial atrophy measurements have shown that CA1 and the subicular atrophy in cognitively healthy individuals is associated with an increased risk of developing MCI, while the gradual involvement of the CA1 and subiculum fields, along with atrophy spread to the rest of the hippocampus (CA2–3 subfields) in amnestic MCI, suggests the future diagnosis of AD [22,23]. Moreover, apolipoprotein E (ApoE) plays a significant role in AD pathogenesis by affecting amyloid and Tau pathology. The presence of the allele ε4 (APOEε4) [24] influences the reduction of the hippocampal volume and the accumulation of Aβ filaments in the brains of elderly people without cognitive impairment and normal levels of Aβ-peptides morphologically [25,26,27]. In this direction, MRI imaging has revealed a significant reduction in the hippocampal volume in amnestic MCI people carrying APOEε4, especially in those who progressed to AD [28]. Even cognitively normal APOEε4 carriers have shown hippocampal volume and cortical thickness reduction together with memory decline and accelerated brain atrophy rates before the onset of cognitive impairment [24,27]. 

Regarding neuronal connectivity dysfunction, a novel PET tracer that binds to synaptic vesicle glycoprotein 2A (SV2A) can be used to quantify synaptic density in vivo, predicting the stage of AD [29]. Several morphological studies have shown that synaptic loss appears early in the pathology [30,31], so the study of markers of neuronal death may derive in promising results for the early diagnosis of AD.

In conclusion, in vivo morphological studies of different brain areas (LC, hippocampus, etc.), along with genetic studies that detect alleles or mutations closely related to the pathology, point out their usefulness as biomarkers for the early detection of AD. Despite the high prognostic ability of these techniques in AD and MCI [32], sometimes, there are limitations that make it difficult to use them in the routine analysis [33]. For this reason, biomarkers in cerebrospinal fluid (CSF) are being extensively studied worldwide as potential candidates for the diagnosis of AD before the appearance of cognitive symptoms [34].

### 2.2. Cerebrospinal Fluid

There is no doubt the ideal fluid biomarker should have a series of characteristics—reliable, reproducible, and noninvasive in terms of collection, specific for a particular disease, simple, and inexpensive to measure, and easy to implement in large populations. In this regard, blood biomarkers meet several of these criteria and could be used in primary care to identify patients with risk of AD [35]. In contrast, CSF collection does not meet the criteria of being a noninvasive procedure, which certainly limits its use but given the close relationship between the brain and the CSF, this fluid could provide valuable information about the biochemical changes that occur in the brain at the preclinical stages of AD [36]. For instance, it is well established that decreased Aβ-42 and elevated total Tau and phospho-Tau in CSF are considered specific markers of AD [37,38], and that these biomarkers can predict cognitive decline over time [39]. The advantages and disadvantages of each category of fluid biomarkers (blood, CSF, and other matrices such as tears, saliva, and urine) are summarized in Table 1.

Nowadays, novel molecular markers are being evaluated in CSF through omics technologies, which allow measuring a large number of analytes at a time (Figure 1). For example, a mass spectroscopy-based analysis revealed that similar levels of ApoE and its isoforms (ApoE2, ApoE3, and ApoE4) were found in the CSF of AD patients and non-AD individuals, independent of their APOE genotype (APOEε2, APOEε3, or APOEε4). However, CSF total ApoE concentrations were positively associated with CSF total Tau and phospho-Tau levels [40,41].

Proteins involved in the pathological processing of the amyloid precursor protein (APP) could be biomarker candidates for early AD diagnosis and must be considered. Presenilin 1 (PSEN1) and β-secretase 1 (BACE1) are both enzymes involved in the cleavage of APP. In MCI patients, both PSEN1 and BACE1 levels and their activity were increased in CSF [42,43]. Moreover, elevated BACE1 expression has been associated with the APOEε4 genotype [43]. It is worth noting that BACE1 activity was only increased in MCI patients whose impairment was progressing to more advanced stages of dementia, and not occurring in stable MCI patients [44]. So, while BACE1 seems to be highlighted as a sensitive early biomarker to detect alterations in the amyloidogenic process in APOEε4 carriers [43], it does not seem to be a good candidate in APOEε4 non-carriers. 

Other early aspects to highlight in AD are neuroinflammation and the synaptic dysfunction; thus, specific markers of these processes could also play a very important role and may correlate more directly with cognitive decline [45]. In this sense, many proteins involved in vesicular transport (secretogranin II (SCG2), chromogranin A (CHGA)), in synapses formation and stabilization (neurexins (NRXNs), neuronal pentraxin 1 (NPTX1), neurocan core protein (NCANP)), and in the immune system (lysozime C (LysC) and β_2_-microglobulin (β_2_M)) were significantly higher in the CSF of patients with MCI, especially in patients with MCI progressing to AD pathology than in AD and healthy control patients [46]. According to one study, higher levels of CHGA in the CSF of healthy elderly people predicted future decreases in Aβ-42 [47]. Other proteins that play a crucial role in inflammation are YKL-40 and visinin-like protein-1 (VILIP-1). Increased expression of these molecules has been seen in both MCI and AD patients, contrary to cognitively normal elderly subjects. While YKL-40 was increased from the prodromal stage until the severe stage of the disease, VILIP-1 was only increased in the prodromal stage [48]. Some studies have found an association between the upregulation of YKL-40 with an increased risk of progression from the normal conditions to MCI [49]. Another potential inflammatory marker is the interferon-γ-induced protein 10 (IP-10), whose level was increased in the CSF of asymptomatic elderly adults that also presented elevated levels of total-Tau and phospho-Tau [50]. Likewise, monocyte chemoattractant protein 1 (MCP-1), a low-molecular-weight cytokine involved in the inflammatory process, was found elevated in the CSF of MCI and AD patients [51]. The triggering receptor expressed on myeloid cells 2 (TREM2), expressed by microglial cells, among others, plays an important role in regulating immune responses in the brain and in the production of inflammatory cytokines [52]. Its haplodeficiency has been associated with increased axonal dystrophy and phospho-Tau accumulation around Aβ-plaques [53]. An increased level of CSF soluble TREM2 has been seen in carriers of an autosomal dominant AD mutation, at least five years before the onset of symptoms, although later to brain amyloidosis and Tau pathology [54]. All these findings reveal that a large number of proteins involved in the inflammatory response can be potential early biomarkers of AD.

One protein that plays an important role in memory enhancement is neurogranin. It is involved in post-synaptic signaling pathways, and its CSF levels differentiated patients with early symptomatic AD from controls with a comparable diagnostic utility to the other CSF biomarkers [55]. The potential of neurogranin as a biomarker of AD depended on the fragment measured [56]. 

Regarding neuronal damage, some proteins such as neurofilament light chain (NF-L), a protein involved in protecting neurites, and neuron-specific enolase (NSE), which plays a role in neuronal metabolism, have revealed increased CSF concentrations in MCI patients in comparison with cognitively elderly, and with patients at advanced AD stages [57,58]. In AD patients with advanced pathology, high CSF NF-L levels are associated with cognitive decline and morphological changes in the brain that indicate neuronal loss [57]. Schmidt et al. showed a correlation between high CSF NSE levels and Tau pathology [58]. These results agree with studies, where plasma protein levels were also studied and support the use of NF-L and NSE as early AD biomarkers [56].

Lipid alterations in CSF participate as well in the modulation of neuropathological events related to AD and can be an AD biomarker candidate. In patients with incipient dementia, a reduction of up to 40% of sulfatide levels was observed [59] The levels of some other lipids such as phosphocholine and sphingomyelin were increased in patients at the prodromal stage and correlated with amyloid and Tau pathology [60]. Another biomarker candidate is the fatty acid-binding protein (FABP3), which may play a role in neuronal synapse formation. In MCI and AD patients, the FABP3 level was higher than in cognitively healthy people [61], and it was related to early structural brain changes typical of AD patients (entorhinal cortex atrophy). Also, high FABP3 levels have been found in non-amnestic elderly APOEε4 carriers [62], showing their increase occurs at a very early stage of the disease [63]. 

Overall, all the findings mentioned above reflect changes in specific areas of the brain detected using novel imaging techniques (MRI, PET), the presence of two classical AD proteins (Aβ and Tau), and the progression of processes such as neuronal apoptosis, synaptic loss, and inflammation. Many of them are still under consideration as potential early biomarkers of AD, and larger longitudinal studies are required for validation of the results [56]. Unfortunately, most AD patients are asymptomatic during the preclinical stages, complicating the recruitment for these kinds of studies and emphasizing the importance of rapid diagnosis.

It is also relevant to investigate the presence of microRNAs (miRNAs), a big family of endogenous short non-coding RNAs that regulate the number of mature mRNAs at the post-transcriptional level [64]. About 70% of identified miRNAs are expressed in the brain, and some miRNAs species are present in exosomes, both good biomarker candidates in clinical diagnostics. Of the approximately 2000 human miRNAs identified to date, no more than 40 are abundantly expressed in the brain [65]. The core CSF biomarkers (Aβ-42, total-tau, and phospho-tau) are relatively stable in clinical AD, and although they are useful for diagnosis, they are not good enough as indicators of disease progression. Although CSF miRNAs are obtained in an invasive manner, which is far from ideal, they have the advantage of targeting important pathological AD genes. A single miRNA has the potential to interfere with the expression of a small family of genes. This is the case with miRNA-125b (upregulated in AD), which targets the synaptic protein synapsin-2 (SYN-2), the enzyme 15-lipoxygenase (15-LOX), and the cell cycle regulator CDKN2A. This opens up the door for the use of miRNAs as therapeutic agents in the future. Furthermore, the misregulation of specific miRNAs could contribute to AD etiopathogenesis [66] and partially explain the large number of brain mRNAs gradually and significantly downregulated in anatomical regions sensitive to AD progression (reviewed in Lukiv 2013)[65]. Other attractive points that favor the use of CSF miRNAs as diagnosis tools are their high stability in body fluids [67,68], the low concentration required for their detection by standard molecular biology techniques, such as quantitative RT-PCR, and the proven high predictive accuracy in the pathogenic process of AD. All of the above supports the huge potential miRNAs offer as diagnostic and prognostic biomarkers and, at the same time, as plausible therapeutic tools against AD [69]. 

In the hippocampus of AD patients, in comparison with healthy volunteers, upregulation of three out of the 13 brain-associated miRNAs studied was observed: miRNA-9, miRNA125b, and miRNA128 [66]. Very similar results were found in the CSF of AD patients. Using microarrays, qRT-PCR and novel highly sensitive LNA, EDC and DIG (LED)-Northern dot-blot (an improved northern blot-based protocol for small RNA detection that combines the use of digoxigenin (DIG)-labeled oligonucleotide probes containing locked nucleic acids (LNA) and 1-ethyl-3-(3-dimethylaminopropyl) carbodiimide (EDC) for cross-linking the RNA to the membrane), high amounts of proinflammatory miRNAs such as miRNA146a, miRNA-155, miRNA-9, and miRNA-125b have been detected in AD patients CSF compared with age-matched controls [65,70,71]. Briefly, miRNA-125b targets synaptic proteins, neurotrophic factors and cell regulator proteins, and miRNA146a targets immune system regulators and proteins involved in proinflammatory signaling, as well as in Aβ accumulation [65]. Both miRNA-125b and miRNA-146a can explain many of the pathogenic effects of AD, so they could be excellent candidates as AD biomarkers. In areas such as the frontal gyrus and the neocortex, the up- and downregulation of an elevated number of miRNAs has been seen, and even some of them showed different regulations according to the area studied [72]. miRNA-29a seems a promising biomarker because it targets BACE1, which promotes the formation of Aβ from APP. In the cortex of AD patients, decreased miRNA29a has been reported, while a two-fold increase in miR-29a levels was found in the CSF of AD patients in comparison with cognitively healthy people [73,74]. The decrease in miR-29 brain expression in AD patients can be associated with an increase of BACE1, leading to the subsequent increase in Aβ levels [75]. Another miRNA that is upregulated in the neocortex, hippocampus, and CSF is miRNA-9 [65,70]. MiRNA-9 is mainly involved in neurogenesis and brain cell proliferation [76,77] and also targets BACE1, decreasing its expression [78]. In this sense, miRNA-29a and miRNA-9 could be indicators of pathology acting as biomarkers. We have only named a few microRNAs involved in AD, although it is worth noting that there are many more with promising results [79].

## 3. Noninvasive Biomarkers

### 3.1. Blood Biomarkers

Blood pressure has been pointed out as an early marker of AD. High blood pressure has been associated with senile plaques, neurofilament tangles, and hippocampal atrophy, and advanced age and hypertension have been linked to AD development [80]. In addition, selected low amounts of brain proteins/substances can cross the blood-brain barrier, reaching the bloodstream. Therefore, it is possible the detection in the blood of specific substances derived exclusively from the brain or systemic pathologies [81]. Despite blood is a more complex matrix for investigating neuronal processes, making the research of neurodegenerative biomarkers in blood challenging; its accessibility makes the study and validation interesting. To this effect, blood represents a noninvasive way of monitoring AD development and progression [82]. 

Compared with CSF, blood is easily collected and, therefore, represents the matrix of choice for the discovery of new accessible biomarkers. In addition, CSF and brain Aβ and Tau correlated with plasma Aβ and Tau in sporadic AD [83,84]. To this end, measurements in the bloodstream of proteins and peptide concentrations that originate in the brain are very promising. A decrease in the levels of Aβ-42, Aβ-40,and the Aβ-42/Aβ-40 ratio was found in the plasma of preclinical AD patients [85]. Other studies focused on Aβ-0 and showed higher levels in samples of AD patients [86]. Blood-based Tau levels have also been investigated in some studies and found to be elevated in the plasma of AD patients [87,88]. However, the relatively low levels of Aβ and Tau proteins in peripheral blood necessitate more sensitive detection techniques to consolidate as diagnostic biomarkers of AD.

Recent studies have revealed that serum neurofilament protein levels correlated with AD [89]. However, this fact was not specific of AD but was also reported in other neurodegenerative diseases [90]. Henceforth, Aβ, Tau, and neurofilaments are not strong enough biomarkers to predict sporadic AD [91], and it would be useful to account for additional molecules for a more accurate early diagnosis. It has been found that changes in the plasma concentration of brain-derived neurotrophic factor (BDNF) depend on the severity of AD [92]. Also, plasma clusterin levels are significantly increased in both, MCI and AD patients [93], and have been related to increased risk of progression from MCI to AD, but a slower cognitive decline in AD patients [94].

Extracellular RNA (exRNA) from human biofluids has been recently characterized. In neurological disorders, brain-derived exRNAs can reach the bloodstream in different ways. One possibility is the elimination of waste from the brain by the lymphatic system into the bloodstream [95]. A second one is that blood-brain barrier leakiness described in early AD facilitates the passage of all types of extracellular molecules [96]. Thus, the presence of exRNAs in the blood allows the study of gene expression in the central nervous system. In this context, it has been recently described that phosphoglycerate dehydrogenase (PHGDH) exhibits consistent upregulation in the AD brain transcriptome and is increased in presymptomatic AD plasma as compared to controls, suggesting the potential utility of plasma PHGDH exRNA as a presymptomatic indicator of AD [97].

Like in CSF, miRNAs are also considered to be one of the potential candidates for blood-based biomarkers. It has been reported that several miRNAs downregulate AD-related proteins, including BACE-1 and APP [98]. Four miRNAs (miR-31, miR-93, miR-143, and miR-146a) were significantly decreased in AD patients’ serum, suggesting that these could be used as potential diagnostic and prognostic markers for dementia. Notably, miR-31, miR-146a, and miR-93 were related to inflammation, cell apoptosis, and fibrosis. Furthermore, miR-93 and miR-146a were significantly elevated in MCI compared to controls and miR-31, miR-93, and miR-146a can be used to discriminate AD from other types of dementia [99]. In addition, the level of miR-206, involved in cognitive decline and memory deficits, was increased in AD plasma, so it could also be a good AD biomarker candidate [100].

Circulating exRNAs are usually protected by exosomes and other extracellular carriers [101]. Extracellular vesicles (EVs), including exosomes, are small (50–150 nm) membrane microvesicles involved in cell-to-cell communication, which can go across a healthy blood-brain barrier. EVs contain not only exRNAs but also other biologically active cargo of molecules specific of their tissue of origin, such as metabolites and proteins, making EVs a good blood-based biomarker candidate, prognostic indicator, and therapeutic tool in AD. Isolation of brain-secreted EVs (BEVs) from the blood provides a minimally invasive way to sample components of brain tissue. Cerebrovascular-derived BEV studies are sparse in human AD patients, so more research would be needed in this field. Nonetheless, it has been reported that pathophysiological alterations in AD are, in fact, reflected in the number and composition of BEVs from neurons, neural precursor cells, and astrocytes [102,103].

As mentioned above, blood is a more complex matrix than CSF, and the high number of cells and soluble molecules contained in it can lead to interferences. Moreover, the low number of brain-derived biomarkers in blood requires highly sensitive techniques for their detection. Great variability in the results depending on the methodology used has been reported, and the fact that several studies show conflicting results represents a limitation for the use of blood biomarkers as an AD diagnostic tool [104]. For all these reasons, the use of blood biomarkers has not yet been validated [81], and the research of alternative fluids as urine, tears, and saliva, is challenging (Figure 2). 

#### 3.1.1. Neuron-Derived BEVs in Blood

Several studies reported significantly elevated Aβ-42 levels in plasma-isolated neuron-derived BEVs in AD dementia relative to cognitively normal individuals. EVs may mediate the transcellular spread of Aβ peptide by destabilizing calcium cell homeostasis and damaging mitochondrial function; thus making neurons more vulnerable to excitotoxicity [105]. Plasma Aβ-42 has the potential to be used both, as a novel earlier biomarkers of AD, as well as a marker of AD progression, having the same capacity as Aβ-42 in CSF for the diagnosis of AD [83]. Nonetheless, the EV concentration is higher in blood plasma than in CSF [105], making this a more sensitive biomarker.

Regarding Tau levels in the plasma, the association of neuron-derived BEVs with AD has yet to reach a consensus. Although three studies [83,106,107] found elevated phospho-Tau levels in AD dementia, these reached a plateau as early as 10 years before AD diagnosis, making it a worse marker of AD progression than Aβ-42 [107]. In addition, three other studies [108,109,110] showed no statistical difference in Tau fragments.

Protein cargo form plasma and serum isolated neuron-derived BEVs included synaptic proteins like neurogranin, synaptotagmin, synaptopodin, and synaptophysin, which was reduced in individuals with AD dementia [111]. However, the potential of being selective biomarkers of AD appears low because these synaptic proteins were also reduced in MCI and Parkinson’s disease. Decreased levels of the growth-associated protein 43 (GAP43), synaptosomal-associated protein-25 (SNAP-25), and synapsin 1 were also observed in blood-isolated neuron-derived BEVs of AD patients [103]. Therefore, synaptic proteins cargo of neuron-derived BEVs demonstrates some biomarker potential in AD, although more studies are needed to confirm this.

Insulin pathway proteins are deregulated in AD as well. Specifically, higher phospho-Ser312-insulin receptor substrate-1 (IRS-1) and lower phospho-panTyr-IRS-1 levels were reported in blood isolated neuron-derived BEVS of AD patients [112,113]. So IRS-1 level, as well as being used to monitor insulin administration [114], could also be used as an AD biomarker. 

Lysosomal proteins of neuron-derived BEVs from plasma were also found to discriminate AD dementia. Levels of cathepsin D, alysosome-associated membrane protein, and ubiquitinated proteins were significantly increased in AD patients, and levels of heat-shock protein-70 were diminished in preclinical and clinical AD, suggesting that neuronal lysosomal dysfunction is an early phenomenon in AD [115].

Finally, research from Winston et al. [106] demonstrated that the level of the repressor element 1-silencing transcription factor (REST) was significantly lower in AD patients and MCI compared to control subjects.

#### 3.1.2. Neural Precursor Cell-Derived BEVs in Blood

Chondroitin sulfate proteoglycan (CSPG4) cells are a subtype of neuronal precursor cells that release neurotrophic factors implicated in neuronal growth and survival. Four assessed neurotrophic factors (hepatocyte growth factor, fibroblast growth factors 2 and 13, and type 1 insulin-like growth factor) were significantly lower in CSPG4 neuronal precursor cells-derived BEVs from preclinical AD patients, being able to use these neurotrophic factors as early biomarkers of AD. No significant further decrease was found during the course of the disease [116], though.

#### 3.1.3. Astrocyte-Derived BEVs in Blood

Astrocyte-derived BEVs have also been reported to cargo Tau, Aβ-42, and APP pathway proteins. However, only levels of BACE1, sAPPβ, complement proteins, and glial-derived neurotrophic factor (GDNF) were significantly deregulated in AD. The levels of BACE1, sAPPβ, and complement proteins were higher [117], and the levels of GDNF were lower in AD patients compared to control individuals [118]. 

#### 3.1.4. MicroRNA Cargo of Blood-Isolated EVs

The levels of miRNAs in peripheral blood can be affected by multiple factors and may also vary among different sample types. In this regard, exosomal miRNAs effectively avoid that problem because of their stable expression. Exosomes are a subtype of EV with a size of 40–100 nm that are released from most types of cells, including neurons [106]. Recent studies have shown that exosomes, in addition to functional proteins, carry mRNA and miRNAs [119], and abnormal expression of these exosomal miRNAs has been detected in AD [120].

More than 40 differentially expressed plasma- and serum-isolated EV-associated miRNAs have been described in AD and MCI relative to control individuals [121,122,123,124,125,126,127]. For example, exosomal miR-342-3p [123], miR-125a-5p, miR-125b-5p, and miR-451a, associated with fatty acid biosynthesis, hippo signaling, and protein processing in the endoplasmic reticulum were significantly lower in AD patients, and their level correlated with the extent of cognitive impairment [128]. Decreased levels of exosomal miRNA 23a-3p, ex-let-7i-5p, ex-miR-126-3p, and ex-miR-151a-3p, which target genes involved in cell death, among others, suggest that changes in the plasma level of AD individuals exhibit diagnostic value [129]. The exosomal miR-223, which regulates inflammation by interacting with different targets, was also significantly decreased in AD patients [125]. On the other hand, Barbagallo et al. found that exosomal miR-29a was significantly increased in AD patients [130] and Cheng et al. reported 14 significantly upregulated exosomal miRNAs [122]. It is also important to mention three of those exosomal miRNAs that have been reported in at least two different studies—the decrease of miR-193b and miR-342-3p [121,124], and both, the increase and decrease of miR3065-5p [122,123]. miR-193b is known to repress the expression of APP and PSEN1 mRNAs, so its reduction may promote amyloidosis, and miR 342-3p is suggested to affect Tau phosphorylation and aggregation.

These studies suggest that specific blood exosome miRNAs can be used as diagnostic biomarkers of AD and, additionally, are able to reflect the disease progression. It has also been reported that the combination of miR-135a, miR-193b, and miR-384, modulators of APP or BACE1 expression, are good for early AD diagnosis [124], demonstrating that a combined biomarker signature is better than a particular one for diagnosis. These studies have been carried out in already clinically diagnosed AD patients, so further studies will be necessary to evaluate the potential of these miRNAs as early biomarkers of AD. Table 2 summarizes all the information regarding AD-related miRNAs found in blood, as well as in CSF.

### 3.2. Ocular Biomarkers

AD not only causes neurodegenerative changes in the brain but also produces structural and functional alterations in the retinal neural and non-neural ocular tissues [151]. Engagingly, specific biomarkers of AD have been reported as well in retinal degeneration and visual function impairment [152], sharing pathophysiological features with glaucoma and age-related macular degeneration (AMD) [153]. The similarity between ocular and cerebral tissues suggests that these ocular manifestations may be used as early biomarkers of AD.

Numerous studies have identified Aβ depositions in the lens and retina. Aβ-accumulates in the retina in an age-dependent manner in a mouse model of AD and correlates with amyloid plaques in the brain. Interestingly, the appearance of retinal amyloid plaques precedes that in the brain [154]. Elevated levels of Aβ1-42 and amyloid plaques were also reported in the retinas of confirmed AD patients [152]. This retinal Aβ can be detected noninvasively by using hyperspectral imaging microscopy or with modified endoscope applied to the corneal surface [155].

In the same way, Aβ has been identified in the lens of rodents, monkeys, and humans in several studies. The accumulation of the isoforms Aβ1-40 and Aβ1-42 has been demonstrated in the lenses of AD people *post mortem* at concentrations comparable with those in the brain [156]. In the same way, a clinical trial carried out on AD patients, and age-matched healthy volunteers confirmed the presence of Aβ in the lens of the first and its correlation in the brain using imaging techniques [157].

Moreover, changes such as thinning of the nerve cell layer, optic nerve atrophy, and the loss of retinal ganglion cells [156,158] were reported in AD, resulting in visual functional impairment and circadian disturbances [159]. Specifically, a study of melanopsin retinal ganglion cells (mRGCs), a photosensitive subtype of ganglion cells in charge of the circadian rhythms, has shown a significant decrease of this neuronal cell type in people with AD but not in healthy controls, with a prominent Aβ accumulation around mRGCs [160,161].

Other ocular tissues, such as the cornea, which is the outermost layer of the eye and, therefore, grants accessibility, could be used as potential biomarker for the diagnosis of AD. Dutescu et al. [162] found the cytoplasmic expression of APP in the epithelial cell layer of the cornea of transgenic mouse models of AD. More recently, Choi et al. corroborated the expression of APP, together with proteins involved in its processing such as BACE1, in human corneal fibroblasts, and the corneal epithelium [163].

For the development of a novel noninvasive screening and diagnostic tool, the ocular examination sector appears promising. In this context, tear fluid provides a viable source widely used for biomarker studies [164], including neurodegenerative diseases [165]. Tear samples are easy to collect and contain a lot of proteins, most notably lipocalin-1, lactotransferrin, and lysozyme C, involved in immune and inflammatory processes [166]. 

Against this backdrop, total protein concentration and composition modifications in tears and an abnormal flow rate and tear function have been described in AD patients [167], supporting the use of tears as a new noninvasive method to discriminate AD patients. Specifically, lipocalin-1, dermcidin, lysozyme C, and lacritin were shown to be potential biomarkers, with an 81% sensitivity and 77% specificity [168]. In addition, the elongation initiation factor 4E (eIF4E) was exclusively expressed in tear samples from AD patients. Total miRNA content was also higher in tears from AD individuals and miR-200b-5p was significantly elevated in AD tear fluid samples compared to controls [148]. Tears could be useful for first screening, and patients with a positive tear analysis test might be further evaluated to establish an early diagnosis. Assessments of pupillary responses and retinal vasculature have also been considered as biomarkers of AD, but are not yet conclusively validated for clinical diagnosis. Further research is needed in order to use ocular biomarkers as AD early diagnostic tools.

Regarding the use of ocular biomarkers, as well as other novel matrices, some limitations arise, like the amall volume of the samples or the standardization of the collection procedures.

### 3.3. Salivary Biomarkers

Due to the link between the decline of the salivary glands and AD [169,170], it seems likely that AD-specific proteins are expressed in the salivary glands. Salivary epithelial cells express APP and Aβ, and changes in the CSF may be reflected in the saliva [171,172]. Saliva is a novel matrix; therefore, there are still some conflicts between studies. One study has revealed no changes in Aβ-40 protein between AD patients and age-matched controls and high levels of Aβ-42 only in MCI patients (in severe AD stages, these levels returned to control values) [173]. A second study has shown that salivary levels of Aβ-40 and Aβ-42 peptides increase as the severity of AD grows, even as far as a three-fold increase in the case of Aβ-42 levels [174,175]. Moreover, studies where the genetic condition was considered, suggested that salivary Aβ-42 levels were associated with familial AD more than with sporadic AD [173,176]. Among the reasons for these different results may be the distinct Aβ peptide detection and saliva collection techniques, as well as the different disease stages of the patients. There is a clear need for more studies and larger sample size to conclude whether there is a relationship between salivary Aβ-42 and Aβ-40 levels and AD progression.

Another typical candidate protein for analysis in the saliva is Tau, which is expressed and secreted by the acinar epithelial cells of the salivary glands [177]. No changes have been seen in total Tau levels between healthy elderly, MCI, and AD patients [178]. In contrast, the phospho-Tau/total-Tau ratio looks promising in this regard. A high phospho-Tau/total-Tau ratio was found in AD patients compared with non-amnestic people [179]. Though no conclusive results were found that pointed in a particular direction, the few results available suggest that studies should be directed at investigating different sites of Tau protein phosphorylation as possible candidates for biomarkers of the pathology [180]. 

Interestingly, the study of one of the most important antimicrobial peptides in saliva, lactoferrin, seems to have a high accuracy in AD diagnosis. Lactoferrin participates in modulating the immune response and inflammation process due to its high defense action. It has been seen that low lactoferrin levels in saliva of healthy people mean a clear risk factor to develop amnestic MCI and AD dementia [181].

Another proposed candidate in the saliva is the acetylcholinesterase enzyme (AChE), which plays a role in removing the accumulation of acetylcholine (ACh) in Aβ-plaques and NFTs. Although there are still few studies, no evidence has been found to suggest that AChE activity is different between AD patients and control individuals of the same cognitive age not taking anticholinergics [182,183].

All these results support the need for longitudinal studies with a larger number of subjects to find conclusive results regarding the potential use of the expression of certain molecules in saliva as biomarkers in AD.

### 3.4. Urine Biomarkers

Oxidative stress plays an important role in AD, and the study of guide molecules of oxidative brain damage might be promising early hallmarks of AD in urine. Some metabolites such as the isoprostane 8,12-iso-iPF(2alpha)-VI, a free amino acid generated by lipid peroxidation whose levels in urine were higher at advanced AD stages, might predict the progression of the pathology from MCI to AD dementia [184,185]. 

Other metabolite candidates that reflect oxidative DNA damage can be oxidized nucleosides such as pseudouridine, 1-methyladenosine, 3-methyluridine, N2, N2-dimethylguanosine, 8-hydroxy-2′-deoxyguanosine, and 2-deoxyguanosine, whose levels were higher in AD patients than in healthy elderly [186].

Other promising markers are some proteins found in urine. Given that the AD-associated neuronal thread protein (AD7c-NTP) was isolated from brain tissue and was increased in the CSF of AD people, correlating with the severity of dementia [187], it has been suggested as a potential biomarker of AD. Recently, a high specificity of this protein has been demonstrated to predict Aβ-plaques in MCI patients when present in the urine [188]. These findings are supported by a meta-analysis proposing the use of urinary AD7c-NTP for the early diagnosis of probable AD [189]. In a longitudinal study carried out in 2018, high levels of albumin, a protein characteristic of chronic kidney disease, were detected in AD patients in comparison with age-matched healthy individuals [190]. According to Yao et al. [191], and in the same direction, the urine of patients with AD showed significantly decreased levels of osteopontin and increased levels of gelsolin and insulin-like growth factor-binding protein 7 in comparison with healthy elderly. All of them are proteins involved in several pathological processes of AD [192,193,194], and they may serve as potential novel urinary biomarkers. 

Although it is evident the need for longitudinal studies with bigger samples to get conclusive results verifying the diagnostic value of these peripheral markers, the scientific community is hopeful that the biomarkers in these noninvasive new matrices will be able to demonstrate diagnostic value.

## 4. Concluding Remarks and Future Perspectives 

In addition to the established Aβ-42, total-Tau, phospho-Tau, and CSF biomarkers, several candidate more accessible fluid biomarkers have shown potential for clinical use in AD to support diagnosis and prognosis.

Blood has been the most widely studied fluid biomarker, being neuron-derived BEVs the most investigated biomarker among blood-isolated EVs. The majority of the studies reported elevated Aβ-42 levels in blood-isolated neuron derived BEVs in AD and MCI patients early in the course of the disease, and also with disease progression. With regard to synaptic proteins assessed in neuron-derived BEVs, growth-associated protein 43 (GAP43) also showed some potential as a marker of AD progression. Henceforth, BEVs have emerged as a novel potential blood-based biomarker of AD. It would also be interesting to study BEVs from other components of the brain, such as the cerebrovasculature, that are affected in the early stages of AD and that would allow us to obtain more sensitive blood-based biomarkers.

Tear fluid also provides a viable source widely used for biomarker studies and lipocalin-1, dermcidin, lysozyme C, lacritin, eIF4E, and the microRNA-200b-5p were shown to be potential biomarkers. The relationship between salivary Aβ-42, Aβ-40, and lactoferrin levels and the progression of AD point out to also be biomarker candidates of AD and MCI in saliva. Lastly, several metabolites and proteins like AD7c-NTP, osteopontin, gelsolin, or insulin-like growth factor-binding protein 7 in urine are involved in several pathological processes in AD.

Inflammatory and oxidative stress markers are very common hallmarks of neurodegenerative diseases such as amyotrophic lateral sclerosis, Parkinson, Huntington, and AD. It has been suggested that these alterations reflect inflammatory mechanisms within the central nervous system that parallel the neurodegenerative process [56]. In this review, we have mentioned several neuroinflammation candidate biomarkers such as TREM2, MCP-1, and YKL-40, which have been extensively investigated in AD patients. Particularly, there is strong evidence regarding CSF YKL-40 levels, not only as a potential biomarker for AD diagnosis, but also as a predictor of disease progression from the asymptomatic stage to prodromal, and eventually dementia stages [51]. However, the idea that these inflammation-related proteins could differentiate AD from other dementias is controversial since neurodegeneration and neuroinflammation go hand in hand and inflammation on its own cannot be considered a marker of a specific pathology. There is a clear lack of reliable inflammatory biomarkers that can be used in the context of accurate diagnosis.

Despite recent studies strongly indicating the potential of fluid biomarkers as early diagnosis of AD, these biomarkers are not yet validated for clinical use and further research is needed before these can be regulatory qualified and applied clinically. Future work should establish normative ranges for the levels of these biomarkers to indicate a pathology that may find clinical applications. If we provided early diagnosis and treatment before the underlying pathology manifests clinically, AD patients’ quality of life could notably improve and could be an approach to prevent its irreversible consequences.

## Figures and Tables

**Figure 1 jpm-10-00114-f001:**
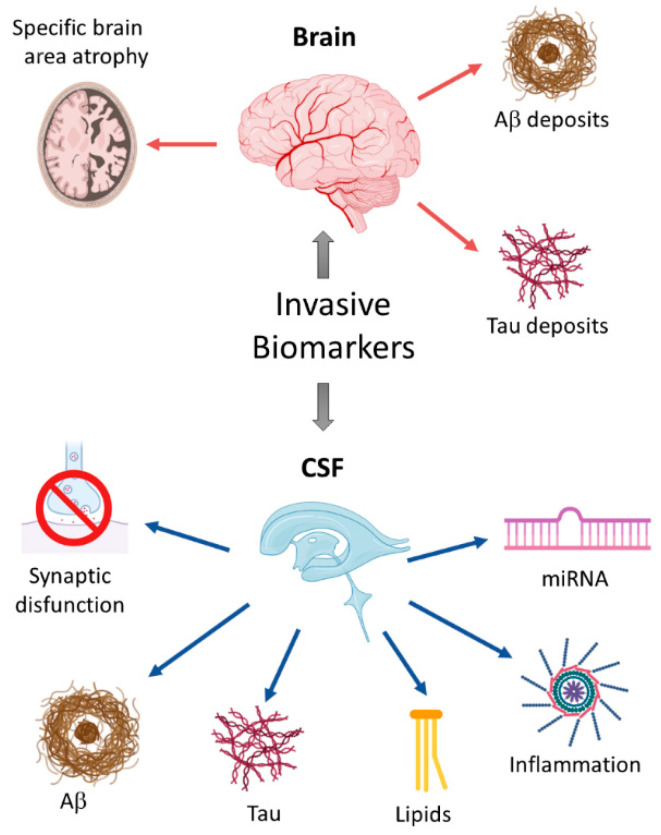
Schematic overview of invasive biomarkers. Different biomarkers have been used to detect early anatomical changes in the brains of people with mild cognitive impairment, including the atrophy of specific brain areas like the locus coeruleus or the hippocampus, and the presence of typical protein aggregates such as extracellular amyloid plaques or intracellular Tau-containing neurofibrillary tangles (upper panel). Additionally, biomarkers of Alzheimer’s disease (AD)-related degenerative processes like synaptic dysfunction, neuroinflammation, oxidative stress, or neuronal loss can be measured in the cerebrospinal fluid of AD patients. The detection of miRNAs represents a novel and promising tool for the early AD diagnosis (lower panel).

**Figure 2 jpm-10-00114-f002:**
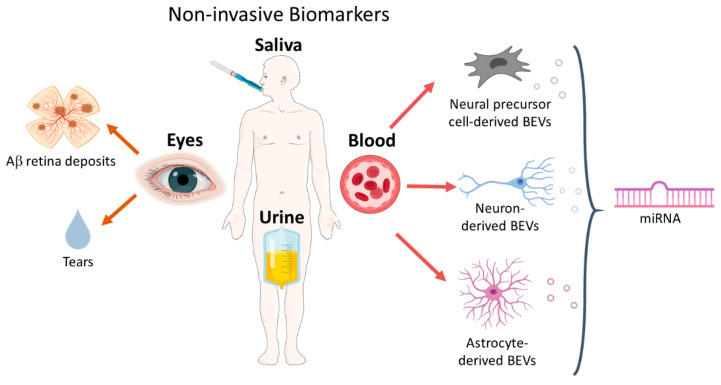
Schematic overview of noninvasive biomarkers: eyes, saliva, urine and blood. Besides fluid biomarkers (tears) that can be collected from the eyes, the promising advances in in vivo retinal imaging could provide an AD diagnosis tool in the near future. Blood-isolated brain secreted extracellular vesicles (BEVs) derive from three possible brain cell types: neural precursors, neurons and astrocytes. The content of this blood-isolated BEVs, mainly miRNAs, have been investigated as potential AD biomarker.

**Table 1 jpm-10-00114-t001:** Advantages and disadvantages for each category of biological fluids used to isolate Alzheimer’s disease biomarkers.

	Advantages	Disadvantages
CSF	Close relationship with the brain High accuracy in the diagnostic process Ability to test a large number of candidate pathophysiological biomarkers High concentration of the biomarkers	Invasive Clinicians require training Positioned in later disease stages, after blood samples, as a confirmatory diagnostic modality Process less accepted by the population and at the risk of causing harm, anxiety, and fear to the patient
Blood	Noninvasive, fast and convenient Inexpensive and reproducible Simple to measure (well-established as part of clinical routines globally) No prior training of the clinicians is required Can be performed in a large variety of settings (primary care, hospitals, patient’s home…) Easy to implement in large populations Ability to test a large number of candidate pathophysiological biomarkers First-step of the multi-stage diagnostic process (identification of patients at the earliest stages of the disease)	Less accurate Presence of very low concentrations of the biomarkers once they have crossed the blood-brain barrier and decreased time window for testing Less consistent results (susceptibility to interference with other components)
Other matrices (tears, saliva, and urine)	Extremely noninvasive Repeatable collections Easy, no risk of infection, can be self-collected by the patient Cheap Stress-free	Remarkable lack of validated studies Lack of results replicated in larger, multicenter and longitudinal studies

**Table 2 jpm-10-00114-t002:** AD-related main miRNA.

miRNAs	Regulation and Localization	References
miR-let-7d-5p, miR-let-7g-5p, miR-26b-5p, miR-191-5p	↓ Blood	[131]
miR-125a-5p	↓ Blood	[128]
miR-126-3p, miR-23a-3p, miR-151a-3p	↓ Blood	[129]
miR-135b	↓ Blood	[132]
miR-181a	↓ Blood	[133]
miR-194-5p	↓ Blood	[134]
miR-19b-3p, miR-29c-3p, miR-125b-3p	↓ Blood	[135]
miR-31, miR-93	↓ Blood	[99]
miR-3613-3p, miR-3916, miR-4772-3p, miR-185-5p, miR-20b-3p	↓ Blood	[123]
miR-501-3p	↓ Blood	[136]
miR-545-3p	↓ Blood	[137]
miR-181c	↓ Blood, ↓ Brain	[133,138]
miR-139-5p, miR-141-3p, miR-150-5p, miR-152-3p, miR-23b-3p, miR-24-3p, miR-338-3p, miR-342-3p, miR-125b-5p, miR-342-5p	↓ Blood, ↓ CSF	[123]
miR-1306-5p	↓ Blood, ↓ CSF	[122,139]
miR-143	↓ Blood, ↓ CSF	[99,133]
miR-15b	↓ Blood, ↓ CSF	[131,133]
miR-15b-3p	↓ Blood, ↓ CSF	[122,139]
miR-193b	↓ Blood, ↓ CSF	[121,124]
miR-223	↓ Blood, ↓ CSF	[125,140]
miR-451a	↓ Blood, ↓ CSF	[128,139]
miR-106, miR-107, miR-181	↓ Brain	[69]
miR-106b	↓ Brain	[138]
miR-137, miR-139, miR-153, miR-183, miR-135, miR-124b	↓ Brain	[66]
miR-15a, miR-19b, miR-26b, miR-330	↓ Brain	[138]
miR-425	↓ Brain	[133]
miR-146b	↓ Brain, ↓ CSF	[133]
miR-210	↓ Brain, ↓ CSF	[133,141]
miR-10, miR-126, miR-127, miR-154, miR-194, miR-195, miR-199a, miR-214, miR-221, miR-338, miR-422b, miR-451, miR-455, miR-497, miR-99a, miR-27a-3p	↓ CSF	[133]
miR-16-2, miR-16-5p, miR-605-5p, mir-9-5p, miR-598, miR-136-3p	↓ CSF	[139]
miR-200b	↓ CSF	[142]
miR-214-3p, miR-299-5p	↓ CSF	[132,143]
miR-29b-3p	↓ CSF	[123]
miR-29c	↓ CSF	[134]
miR-29	↓ Blood, ↓ Brain, ↑Brain	[69,131,133]
miR-125b	↓ Blood, ↑ Brain, ↑ CSF	[65,66,123]
miR-146a	↓ Blood, ↑ Brain, ↑ CSF	[69,71,99]
miR-26a	↓ Brain (frontal cortex), ↑ Brain (hippocampus)	[133]
miR-3065-5p	↓ Blood, ↑ Brain	[122,123]
let-7i-5p	↓ Blood, ↑ CSF	[129,134]
miR-106a-5p, miR-20-5p, miR-425-5p, miR-18b-5p, miR-582-5p	↑ Blood	[122]
miR-106b-3p, miR-20b-5p, miR-146a-5p, miR-195-5p, niR-497-5p	↑ Blood	[135]
miR-455-3p, miR-4668-5p	↑ Blood	[144]
miR-5001-3p	↑ Blood	[123]
miR-519	↑ Blood	[140]
miR-548at-5p	↑ Blood	[123]
miR-590-5p	↑ Blood	[134]
miR-101-3p, miR-106b-5p, miR-143-3p, miR-335-5p, miR-361-5p,	↑ Blood, ↑ CSF	[122]
miR-138-5p	↑ Blood, ↑ CSF	[123]
miR-155	↑ Blood, ↑ CSF	[71,131]
miR-15a-5p	↑ Blood, ↑ CSF	[122,134]
miR-659-5p	↑ Blood, ↑ CSF	[123]
miR-100, miR-145, miR-148a, miR-27, miR-34a, miR-381, miR-422a, miR-423, miR-92	↑ Brain	[133]
miR-128	↑ Brain	[66]
miR-34	↑ Brain	[69]
miR-98	↑ Brain	[138]
miR-let-7b, miR-let7e	↑ CSF	[145]
miR-let-7f, miR-105, miR-138, miR-141, miR-151, miR-186, miR-191, miR-197, miR-204, miR-205, miR-216, miR-302b, miR-30a-3p, miR-30a-5p, miR-30b, miR-30d, miR-32, miR-345, miR-362, miR-371, miR-374, miR-375, miR-380-3p, miR-429, miR-448, miR-449, miR-494, miR-501, miR-517, miR-518, miR-520, miR-526	↑ CSF	[133]
miR-20a-5p	↑ CSF	[122]
miR-222	↑ CSF	[146]
miR-331-5p, miR-485-5p, miR-132-5p	↑ CSF	[139]
miR-613	↑ CSF	[147]
miR-200b-5p	↑ Eyes	[148]
miR-93-5p	↑ ↓ Blood, ↑ CSF	[122,135]
miR-101	↑ Blood, ↓ Brain	[131,138]
miR-132, miR-212	↑ Blood, ↓ Brain	[126,133]
miR-200c	↑ Blood, ↓ Brain (frontal cortex), ↑ Brain (hippocampus)	[133,149]
miR-9	↑ Blood, ↓ Brain (frontal cortex, cortex), ↑ Brain (hippocampus), ↑ CSF	[66,71,131,133,138]
miR-30e-5p	↑ Blood, ↑ Brain, ↑ CSF,	[122,133]
miR-29a	↑ Blood, ↓ Brain, ↑ CSF	[73,74,130]
miR-206	↑ Blood, ↑ CSF, ↑ Eyes	[100,150]
miR-142-5p	↑ Blood, ↓ CSF	[133,134]
miR-384	↑ Blood, ↓ CSF	[124]
miR-135a	↑ Blood, ↓ CSF, ↑ CSF	[124,133,142]
miR-125a	↓ Brain, ↑ CSF	[66,133]
miR-29b	↓ Blood, ↓ Brain, ↑ CSF	[73,74,131]
miR-30c	↑ Brain (frontal cortex), ↓ Brain (hippocampus), ↑ CSF	[133]

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
