# Peer review of "Biomarkers for Alzheimer’s Disease Early Diagnosis"

_jpm, 2020, doi:10.3390/jpm10030114_

Round 1

Reviewer 1 Report

Ausó and colleagues discuss biomarkers for the early diagnosis of Alzheimer’s disease. This is a very important and timely topic. However, the reading of the paper is not fluid and the language also to be revised.

Some other suggestions are provided to improve the paper:

The authors discuss several proteins (e.g. involved in inflammation) and miRNAs that have been observed in MCI and AD patients brains and CSF. Considering that these molecules are obtained in an invasive manner, what is the advantage of evaluating their levels in comparison to the traditional biomarkers (e.g. phospho- and total tau and Abeta42 in CSF)?

Considering miRNAs, authors indicate that several other than those mentioned in the paper are being studied as potential biomarkers. It would be interesting to include a table with all these miRNAs.

Concerning the biomarkers evaluated in peripheral blood, authors must also include papers showing negative results i.e. that do not support the use of peripheral blood biomarkers, and discuss the value of these molecules as biomarkers for AD. The same for ocular biomarkers.

Authors should also debate the use of inflammatory and oxidative markers for early AD diagnosis considering that these markers also appear in several other diseases.

The legends of the figures must be self-explaining.

Reviewer 2 Report

In this paper, the authors gave a comprehensive overview of biomarkers in the early diagnosis of Alzheimer’s disease in different biological fluids. I think that the manuscript is very interesting and of a great interest in both clinical and research fields. However, I have some suggestions to improve its quality:

- A resuming table including main advantages and disadvantages for each category of biological fluids should be useful for the reader to immediately make a comparison among biomarkers.

- In addition to miRNAs, have other non-coding RNAs been also investigated as potential biomarkers?

- The authors should report if the most promising biomarkers are specific for early AD or/and other neurodegenerative disorders.

- Authors should cite “Galimberti. J Alzheimers Dis. 2014; 42(4):1261-7” in the miRNA’s section.

- English needs to be carefully checked for a native speaker.

Round 2

Reviewer 1 Report

The paper has been significantly improved.